# Interplay between electrochemical reactions and mechanical responses in silicon–graphite anodes and its impact on degradation

Junhyuk Moon [1,8 ✉], Heung Chan Lee [1,8 ✉], Heechul Jung[1,5], Shinya Wakita[1,6], Sungnim Cho[1,6], Jaegu Yoon[1,6], Joowook Lee[1,6], Atsushi Ueda[1,7], Bokkyu Choi[1,6], Sihyung Lee[1], Kimihiko Ito[2], Yoshimi Kubo [2], Alan Christian Lim[3], Jeong Gil Seo[3], Jungho Yoo[4], Seungyeon Lee[1], Yongnam Ham[1], Woonjoong Baek[1], Young-Gyoon Ryu [1,6 ✉] & In Taek Han[1]

Durability of high-energy throughput batteries is a prerequisite for electric vehicles to penetrate the market. Despite remarkable progresses in silicon anodes with high energy densities, rapid capacity fading of full cells with silicon–graphite anodes limits their use. In this work, we unveil degradation mechanisms such as $Li^+$ crosstalk between silicon and graphite, consequent $Li^+$ accumulation in silicon, and capacity depression of graphite due to silicon expansion. The active material properties, i.e. silicon particle size and graphite hardness, are then modified based on these results to reduce $Li^+$ accumulation in silicon and the subsequent degradation of the active materials in the anode. Finally, the cycling performance is tailored by designing electrodes to regulate $Li^+$ crosstalk. The resultant full cell with an areal capacity of 6 mAh $cm^{-2}$ has a cycle life of >750 cycles the volumetric energy density of 800 Wh $L^{-1}$ in a commercial cell format.

[1] Samsung Advanced Institute of Technology, Suwon-si, Gyeonggi-do, Korea. [2] C4GR-GREEN, National Institute for Materials Science, Tsukuba, Ibaraki, Japan. [3] Department of Chemical Engineering, Hanyang University, Seongdong-gu, Seoul, Korea. [4] National NanoFab Center, Daejeon, Korea. [5] Present address: Department of Energy and Mineral Resources Engineering, Dong-A University, Bumin Campus, 225, Seo-gu, Busan, Korea. [6] Present address: Samsung SDI, Suwon-si, Gyeonggi-do, Korea. [7] Present address: Asahi Kasei Corporation, Fuji-shi, Shizuoka, Japan. [8] These authors contributed equally: Junhyuk Moon, Heung Chan Lee. ✉email: jh.d.moon@samsung.com; hchan.lee@samsung.com; ygryu@samsung.com

In 2018, more than 21% of US greenhouse gas (GHG) emissions were from automobiles[1,2]. Therefore, the electrification of automotive transportation will have a significant impact on reducing dispersed GHG emissions. For successful penetration of the petroleum-based transportation market, electric vehicles (EVs) must have batteries with high energy throughput and good durability[3]. EV batteries are widely expected to require energy densities of more than 800 Wh L$^{-1}$ at cell level to achieve driving ranges greater than 500 km and a cycling performance of 1300 cycles[3–5].

Silicon is a candidate anode material for achieving the required energy density of EV batteries because of its high theoretical specific capacity. However, silicon anodes suffer from rapid capacity degradation owing to large volume changes during cycling, which lead to pulverisation of the anode, loss of electrical contact, and instability of the solid electrolyte interphase (SEI). Previous efforts to improve the cycling performance of silicon anodes have primarily focused on resolving the rapid fading issues by using nanostructured silicon[6–9] or carbon-coated/mixed silicon particles[10–14], or by forming a stable SEI[15–18]. Indeed, some half cells have exhibited outstanding performance over thousands of cycles. However, the full-cell performance is not directly related to the half-cell performance for two fundamental reasons. First, the amount of Li$^+$ in a full cell is limited by the cathode capacity, whereas it is practically unlimited in a half cell. Therefore, half cells do not capture the issue of Li$^+$ loss during the formation stage (the initial charge/discharge cycling to form the SEI) due to low initial Coulombic efficiency of the silicon anode. Second, the potential of the anode in a full cell is regulated by the potential difference between the cathode and anode, which is fundamentally different from the potential of the anode versus a thermodynamic reference potential during cell operation.

It is necessary to understand the detailed interactions between silicon and graphite in silicon–graphite composite anodes, such as the (de)lithiation kinetics, electrochemical behaviour, mechanical response, and their interplay with cycling stability[19]. Identifying the origin of long-term degradation in full cells is particularly important for improving the cycling performance of complete batteries with silicon–graphite anodes. Up to now, despite efforts to develop reliable analytical techniques that can identify the Li$^+$ content in Li–Si alloys[20–24], amorphous nature of lithium silicide that does not produce diffraction peaks has hindered investigations of the interaction between silicon and graphite during full-cell operation; its visualisation has also been challenging because of the complexity of the experimental environment[25]. Moreover, the silicon content in commercial silicon–graphite anodes has thus far been limited to 2–3 wt% to guarantee industrially acceptable cycle numbers[3,26–28], yet to realise EVs with driving ranges that can compete with those of fossil fuel vehicles, the anodes are required to have a 7 wt% silicon content, even when using cathode materials with high specific capacities[29].

In an attempt to unravel the phenomena underpinning the long-term degradation of silicon anodes and fulfil commercial requirements, we design and implement a full cell using a silicon–graphite anode and quantitatively characterised the states of each component (silicon and graphite) through *operando* X-ray diffraction (XRD) analysis. In this work, we observe the following phenomena: Li$^+$ crosstalk between the silicon and graphite particles; Li$^+$ accumulation inside the silicon core during cycling; and capacity depression of graphite by pressure-induced staging transition[30–34] due to gradual expansion of the silicon particles. To the best of our knowledge, these phenomena, which are closely related to the mechano-electrochemical relationship between silicon and graphite, have never been reported before, even in half cell studies. Here, the focus is on the relationships among these phenomena and their impact on long-term anode

degradation. By clarifying the proposed degradation mechanism, we are able to relieve the long-term detrimental effects, and achieve a cycling performance of >750 cycles in a prismatic cell with a high capacity of 8.7 Ah.

## Results

**Li$^+$ trapping in silicon due to Li$^+$ crosstalk between silicon and graphite.** In the formation stage, a stable interface (the SEI) forms between the active material and electrolyte in a cell. During this process, the silicon transforms into an amorphous phase which is invisible to diffraction[35]. Thus, it is challenging to quantify the Li$^+$ content in the silicon and graphite particles. Herein, we present an analytic basis for Li$^+$ quantification by *operando* XRD. By considering the XRD data for a full cell (denoted as Cell A) containing a graphite-only anode paired with a high-nickel cathode, Li$_{1.0}$Ni$_{0.88}$Co$_{0.08}$Mn$_{0.04}$O$_2$, the amounts of Li$^+$ in silicon and graphite in a second full cell (denoted as Cell B) containing a silicon–graphite anode can be determined (see Supplementary Methods 1, Supplementary Note 1, and Supplementary Fig. 3 for details). Figure 1a shows the Li$^+$ content of the graphite and silicon particles of the composite anode in Cell B during cycling, consisting of constant current (CC) and constant voltage (CV) charging steps, a rest step, and a CC discharging step.

Figure 1b compares the state-of-charge (SOC) of the individual components (hereinafter, individual SOCs) from the XRD data of a full cell at 0.5 C (circles in Fig. 1b) with the values calculated from electrochemical analysis of half cells (solid lines in Fig. 1b). Sudden discontinuous changes appear in the experimental individual SOCs of silicon and graphite, marked by red triangles in Fig. 1b. These changes correspond to phase transitions of Li$_x$Si, which cause abrupt changes in the potential[36,37] (see Supplementary Note 2 for details). Given the two-phase lithiation process[38–40], we can determine the Li$_x$Si phase in the outer part of the silicon particles, known as the shell. Moreover, the large deviation between the experimental and calculated individual SOCs towards the end of the charging step indicates Li$^+$ crosstalk between the silicon and graphite. The starting point of this deviation is marked by a blue triangle in Fig. 1b. By 81.7% SOC, 9.4% of the total amount of Li$^+$ is exchanged from silicon to graphite. Because of the anode utilisation, the graphite can accommodate these amounts of Li$^+$. Although similar electrochemical phenomena, such as crosstalk among active materials, have been reported for internal redox couples in blended cathodes[41], they have not yet been reported for anodes, because there have been no thorough in situ analyses of Li$^+$ in anodes of full cells during the CV and rest steps of the charging process.

Schematics of the active materials at various SOCs are given in Fig. 1c based on the *operando* XRD results. There is a significant transfer of Li$^+$ from silicon to graphite between points iii and iv in Fig. 1c, which corresponds to the rapid decrease in the silicon SOC and sudden increase in the graphite SOC around 80% SOC in Fig. 1b. Herein, point iii corresponds to the maximum Li$^+$ content in silicon. The silicon SOC decreases immediately after the maximum point because of the third phase transition of silicon, indicated by the red triangle at 65.4% SOC in Fig. 1b. Within a wide capacity range, the potential of silicon is higher than that of graphite (Supplementary Fig. 4); thus, Li$^+$ fills silicon first during charging of the full cell. However, there is little movement of Li$^+$ in the shell of the silicon particles into the inner core because of the pressure caused by volume expansion of the lithiated shell. This creates a high degree of inhomogeneity within the silicon particles, which has previously been reported as a cause of a large kinetic resistance and the origin of potential hysteresis of silicon[42]. Therefore, graphite particles in contact with the silicon shell might collect the surface-localised Li$^+$ potential of silicon particles.

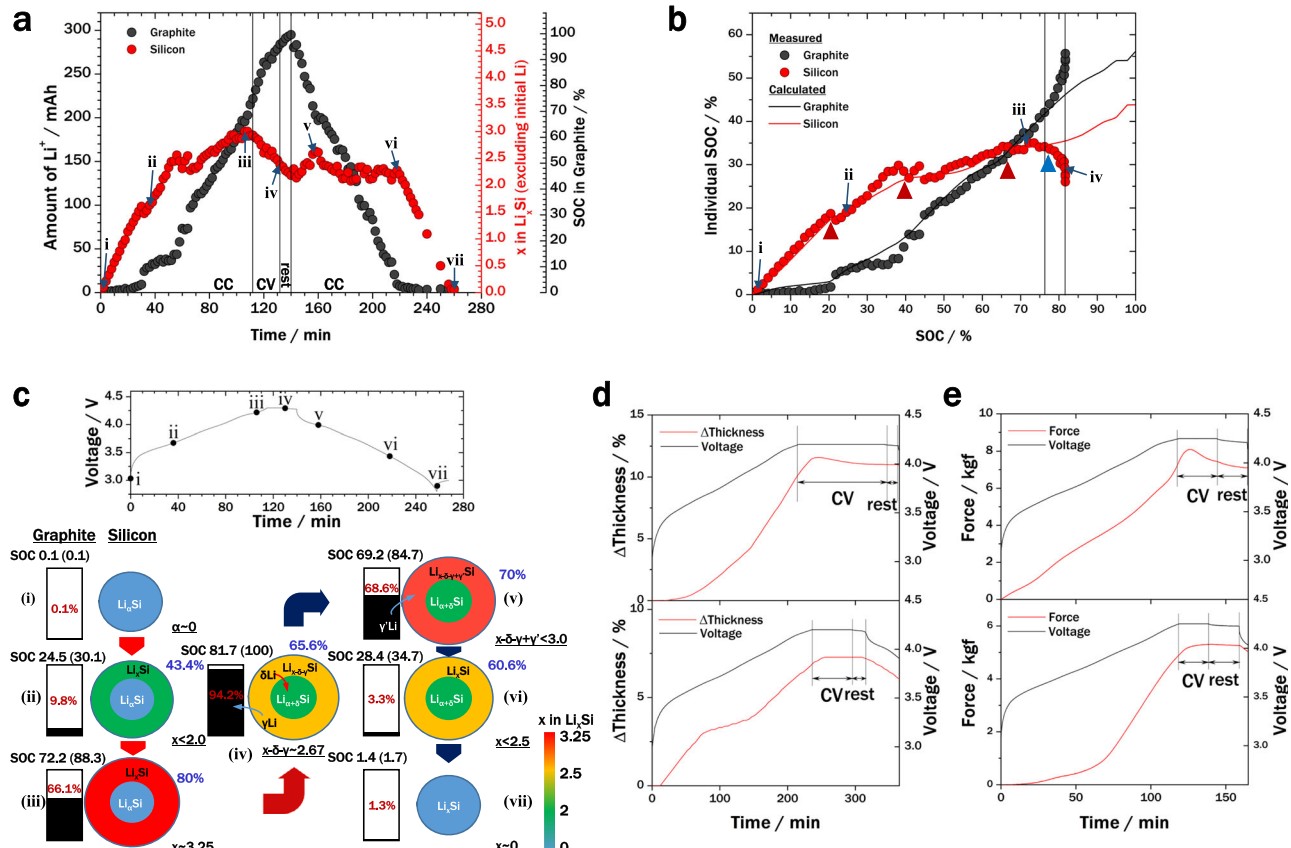

**Fig. 1 Deconvolution of the mixed state of Cell B at 0.5 C. a** Deconvolution of the mixed state of the anode of Cell B into individual silicon and graphite states. **b** Amount of $Li^+$ in the calculated and experimental (lines and circles, respectively) states-of-charge (SOCs) for individual components. The discrepancies at high SOCs originate from the utilisation of the active materials and the negative to positive ($n/p$) ratio. **c** Voltage profile (upper panel) and schematics of active materials at various anode SOCs (lower panel). The letters i–vii correspond to the points in (**a**) and (**c**) (upper panel). The numbers in parentheses represent the full-cell SOCs, and the corresponding graphite and silicon SOCs are indicated in red and blue, respectively. **d** Thickness variations (Δthickness; red line) and voltage profiles (black line) of the cells using silicon–graphite (upper panel) and graphite (lower panel) during constant current (CC)–constant voltage (CV) charging and rest mode at a C-rate of 0.25 C. The charging start time is set to 0 min. **e** Evolution of forces on the pouch cells (red line) and corresponding voltage profiles (black line) of cells using silicon–graphite (upper panel) and graphite-only (lower panel) anodes during CC–CV charging and rest mode at 0.5 C.

Given that the chemical potential of each component is altered by the change in volume owing to applied pressure, the chemical potentials of the silicon particle shell and graphite can be calculated (red and black circles, respectively, in the upper panel of Supplementary Fig. 5). There is a gradual relaxation of the potential difference (Δμ, blue circles in Supplementary Fig. 5) during the CV charging and rest steps, which explains the occurrence of the $Li^+$ crosstalk (see Supplementary Note 4 and Supplementary Fig. 5 for details). The internal redox reaction between silicon and graphite, which was indicated by the XRD results, was also evidenced by changes in the thickness of the cell during the CV charging and rest steps, as shown in Fig. 1d. Furthermore, the compression force on the cell decreased, as shown in Fig. 1e (see Supplementary Note 6 for details). A time delay was observed in both measurements, which has also been reported previously[43,44].

The thickness measurements were performed using "pouch" cells, which were fabricated by folding the cell stack in half lengthways, with the cathode on the inside. The pouch provides space for evolved gases to escape. However, the in situ thickness measurements were affected by rearrangement of the active materials and viscoelastic components such as the pouch, separator, and porosity in the electrodes. By considering only the volume change of the active materials, the cell expansion can

be calculated from Fig. 1a using the volumetric strains of the active materials;[22,45,46] under this assumption, the difference in cell thickness between the start and end points of $Li^+$ crosstalk during the CV and rest steps would be 1.8%. However, the measured thickness difference is 0.65% (Fig. 1d). This indicates that rearrangement of the active particles and mechanical damping by the viscoelastic components occur.

It should be noted that the complete battery cell environment in which this crosstalk phenomena is observed cannot be reproduced in electron microscopy techniques, as additional specimen preparation steps are required, which often leave cleanly cut surfaces. Thus, only bulk and straightforward observations of full battery cells are discussed herein. The limitations of microscopic visualisation are briefly discussed in Supplementary Note 18 (also see Supplementary Figs. 23–26).

$Li^+$ movement from the shell to the core of silicon particles is hindered by the increasing stress during anode lithiation in the full cell[47]. However, the stress applied to the silicon core is alleviated during the CV charging step because of $Li^+$ crosstalk, as $Li^+$ transfers from the silicon particle shell to the graphite particles. At the same time, a portion of $Li^+$ in the shell can penetrate into the depressurised core of the silicon particle, as depicted in the schematic illustration of point iv in Fig. 1c. This is explained by considering the effect of pressure in two-phase

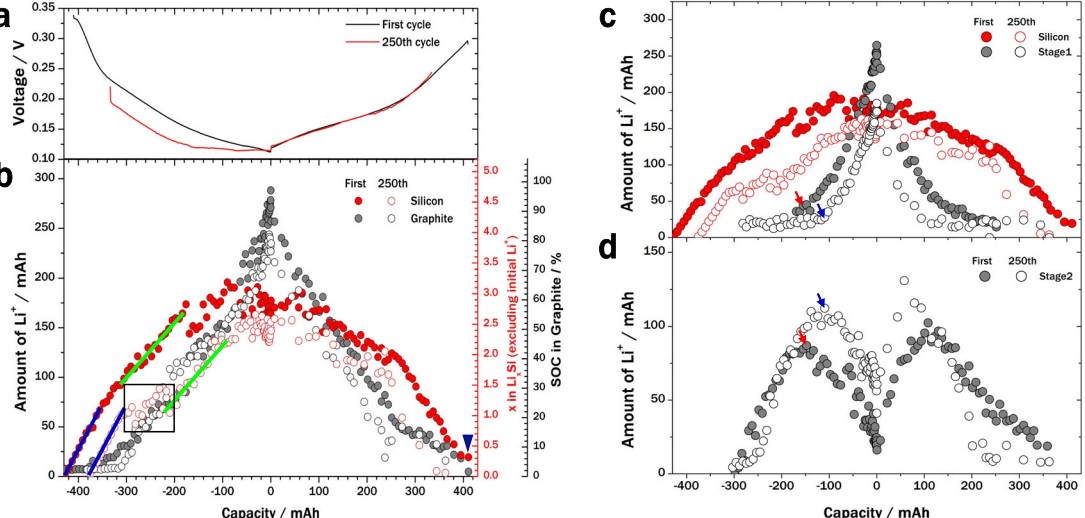

**Fig. 2 Li+ accumulation in silicon. a** Anode potential determined by three-electrode cell measurements during cycling at a C-rate of 1C. At the start of charging, the open-circuit voltage (OCV) of the 250th cycle (red line) is lower than that of the first cycle (black line). **b** Amounts of Li+ in silicon (red circles) and graphite (black circles) for the first (filled circles) and 250th cycles (open circles) compared to the state-of-charge (SOC) in graphite. There is a delay in Li+ transfer to silicon, indicated by the black box. At the end of discharging in the first cycle, some Li+ remains in silicon (blue triangle). **c** Amounts of Li+ in silicon (red circles) and graphite stage 1 (black circles) for the first (filled circles) and 250th cycles (open circles). **d** Amounts of Li+ in graphite stage 2 for the first and 250th cycles (filled and open black circles, respectively).

lithiation of a spherical silicon particle[39] (see Supplementary Note 7 for details of the analytic calculation). It is hard for this penetrated Li+ to delithiate from the silicon during the next discharge step of the full cell owing to the increased distance from the surface. Thus, a higher C-rate results in more Li+ remaining in the silicon at the end of the first cycle (Supplementary Fig. 6). This trapping of Li+ in the silicon core is similar to the two-way diffusion trapping model[48].

**Long-term degradation originating from Li+ accumulation in silicon.** During long-term cycling, the full-cell performance degrades, with a loss of capacity. The left-hand sides of the curves in Fig. 2a represent the open-circuit voltages (OCVs) of the cell before the first and 250th cycles (0.34 and 0.22 V, respectively). This decrease in OCV after long-term cycling may be related to the accumulation of Li+ in the core of the silicon particles. From the potential profile of the silicon–graphite anode at a slow C-rate (blue line in Supplementary Fig. 4), the amount of Li+ in the anode at the OCV before the first and 250th cycles was estimated to have increased by 11.2%. This increase agrees well with the value determined by inductively coupled plasma atomic emission spectroscopy (10.8%; see Supplementary Note 8, Supplementary Table 1, and Supplementary Figs. 8–10), considering OCV measurements reflecting various types of cell resistances. The loss of Li+ owing to the formation of the SEI mostly occurs during the formation stage; during subsequent cycling, less Li+ enters the SEI ("inactive lithium", red filled circles in Supplementary Figs. 9 and 10) than the amount of Li+ that remains in the anode ("remaining Li+, blue filled circles in Supplementary Figs. 9 and 10) after to 250 cycles. The gradual increase in the SEI was additionally confirmed by X-ray photoelectron spectroscopy (XPS), electron probe micro-analysis (EPMA), and direct current–internal resistance (DC–IR) measurements, and discussed in detailed in Supplementary Notes 10, 11, and 15, respectively (also see Supplementary Figs. 13, 14, and 20). The amounts of Li+ remaining in the silicon and graphite particles at the end of a cycle at 1 C (blue triangle, Fig. 2b) were 19.4 and 4.7 mAh, respectively, and these remaining amounts of Li+ contributed to a 3.7% loss in the charging capacity.

The amount of Li+ remaining in silicon gradually increases, with one inflection point observed at −350 mAh for the first cycle during full-cell charging from −400 to −200 mAh (red filled circles, Fig. 2b). In this range, the sudden changes in the amount of Li+ are less distinct owing to the C-rate capability of silicon (see Supplementary Note 12 and Supplementary Fig. 15). However, two inflection points were observed for the amount of Li+ in silicon at the 250th cycle (red open circles, Fig. 2b). The amount of Li+ in silicon increases with the same slope (blue line) as in the first cycle until the first inflection point (from $x = 0$ to 1 in Li$_x$Si). From the second inflection point, the amount of Li+ in silicon again increases with the same slope (green line) as in the first cycle (from $x = 1$ to 2.2 in Li$_x$Si). The delay region for Li+ transfer (indicated by the black box in Fig. 2b) could suggest that Li+ is slowly inserted into the silicon particles owing to the presence of accumulated Li+ in the inner core of the silicon particles. Deviation from the expected Li+ insertion behaviour, as calculated from the electrochemical measurements, is shown in Supplementary Fig. 16 (see Supplementary Note 13 for details).

The OCV differences, residual Li+ contents according to the XRD analyses at the first and 250th cycles, and delayed Li+ insertion profiles after 250 cycles all corroborate the phenomenon of Li+ accumulation in the inner core of the silicon particles during cycling. This Li+ accumulation was also visualised through scanning transmission electron microscopy-electron energy loss spectroscopy (STEM-EELS) (see Supplementary Note 9 and Supplementary Figs. 11 and 12). In addition, the limited increase in the internal resistance during cycling (Supplementary Fig. 20) further supports the accumulation of Li+ in the inner core. It should be noted that Li+ accumulation inside the silicon core would lead to irreversible capacity fading; the increase in the inner volume of the silicon particle would cause mechanical stress on the outer shell, and sequential chemical reactions of the silicon surface with the electrolyte and degradation of the silicon surface would be accelerated.

Figure 2c and d show the amount of Li+ in silicon and in each graphite stage (LiC$_6$ and LiC$_{12}$, denoted as stage 1 and 2, respectively) at the first and 250th cycles (filled and open circles, respectively). The amount of Li+ at graphite stage 1 in the first

cycle (black filled circles, Fig. 2c) is larger than that in the 250th cycle (black open circles, Fig. 2c), whereas the amount of Li$^+$ at graphite stage 2 in the first cycle (black filled circles, Fig. 2d) is smaller than that in the 250th cycle (black open circles, Fig. 2d) from −200 to 200 mAh. Simultaneously, Li$^+$ accumulation in the silicon core during cycling results in the expansion of the silicon, which increases the pressure on graphite during full-cell charging. This phenomenon was verified by the pressure evolution calculated from the stage 1 peak shifts during full-cell charging (Supplementary Figs. 17 and 18, and Supplementary Note 3): during the first cycle, the pressure on graphite continuously increases; however, after 250 cycles, the pressure on graphite steeply increases before CV charging.

The onsets of Li$^+$ in stage 1 of graphite arise at −154.69 mAh (corresponding time: 39 min) and at −109.44 mAh (38 min) for the first and 250th cycles, respectively (blue and red arrows in Fig. 2 c and d). At these points, the graphite is experiencing 0.290 and 0.329 GPa of pressure in first and 250th cycles, respectively. At the points of maximum lithiation in silicon (−83.02 mAh (49 min) for first cycle and −28.96 mAh (53 min) for 250th cycle), the pressures on graphite increase to 0.295 and 0.388 GPa for first and 250th cycles, respectively, as shown in Fig. 2c and Supplementary Fig. 18. The increase in pressure impacts the lithiation of graphite. The critical pressure of graphite for the pressure-induced staging transition would be ca. 0.33 GPa. Previous studies on pressure-induced staging transitions under several hundred MPa have been reported[30–34]. Consequently, the transition from graphite stage 2 to stage 1 is delayed, as indicated by the belated onset of Li$^+$ in stage 1, as well as the shift in the maximum of stage 2 after the 250th cycle (blue arrows, Fig. 2c and d). This behaviour explains the capacity fading of the graphite component of the anode during long-term cycling. The capacities of silicon and graphite at the end of charging were 158.7 and 275.1 mAh in the first cycle, respectively, yet just 144.4 and 235.6 mAh in the 250th cycle, respectively (Supplementary Fig. 19). This equates to a charge capacity retention in silicon and graphite of 91.0% and 85.6%, respectively. Thus, the degradation in graphite is 60% more severe than that in silicon.

Here, the rapid volume expansion and increase in internal resistance caused by side reactions at the silicon/electrolyte interface were not observed, which might have been due to the nature of the microscopic structure of the surface-treated silicon/carbon (SSC) composite particles used in the anode, whereby the silicon particles are not in direct contact with the electrolyte (initial Coulombic efficiency: 86.4%, the capacity retention at 200th cycle: 86.6%, see Supplementary Fig. 1 and Supplementary Note 9 for electrochemical properties and SEI formation on SSC, respectively). For this reason, the previously reported total breakage of graphite owing to excessive volume expansion of silicon[23,49] was not observed.

**Improvements over existing materials and electrode designs.** With a better understanding of the phenomena of Li$^+$ crosstalk and accumulation inside the silicon particles, it is possible to improve the existing materials and electrode design to reduce the degradation of the cell. For example, reducing the size of the silicon particles would shorten the transport length of Li$^+$ and aid its transfer out of the silicon particles during discharging, thereby preventing Li$^+$ accumulation in silicon due to Li$^+$ crosstalk. Figure 3a and d show cross-sections of SSC composites fabricated using silicon particles with long-axis lengths of ~100 and ~85 nm (denoted as 100- and 85-nm Si), respectively. Table 1 summarises the average dimensions of the silicon particles in the SSCs, as obtained by image analysis (see Supplementary Note 14). These

silicon particles are smaller than the critical size for silicon particles in anode materials (sphere: ~150 nm; cylinder: ~250 nm), so they are not expected to experience mechanical fracture on lithiation[6–9].

The initial Coulombic efficiencies measured for silicon–graphite anodes using 85- and 100-nm Si were 88.9 and 90%, respectively. The slightly lower value for 85-nm Si might be due to increased surface reaction. In the total-electron-yield (TEY)-X-ray absorption near edge structure (XANES) signals of pristine electrodes, the silicon oxide peak of 85-nm Si is larger than that of 100-nm Si, as shown in Supplementary Fig. 21. As expected, reducing the silicon particle size alleviates Li$^+$ accumulation in the inner core because of the shorter transport length of Li$^+$ in silicon for (de) lithiation. Over long-term cycling, this would decrease silicon expansion and surface degradation. Figure 3b and e show TEY-XANES spectra that represent the degradation of the silicon surfaces of the 100- and 85-nm Si particles during full-cell cycling. The peaks at 1,844 and 1,847 eV correspond to lithium silicate (Li$_x$SiO$_y$) and silicon oxide (SiO$_x$), respectively, which lead to high impedance and low capacity retention[50–52]. The TEY-XANES signals at 1,844 and 1,847 eV are stronger for the cell using 100-nm Si (Fig. 3b), which verifies that surface degradation of the silicon particles in this cell is more severe than in the cell using 85-nm Si (Fig. 3e) after 50 cycles. By contrast, partial-fluorescence-yield (PFY)-XANES spectra, which characterise the bulk properties of the silicon particles, show little change during cycling for 100- and 85-nm Si (Fig. 3c and f, respectively), unlike the evolution of PFY-XANES spectra over cycling for a cell wherein silicon is in contact with the electrolyte[53].

We also carried out in situ thickness measurements of pouch cells during cycling (Fig. 3g). The increase in the overall thickness of the cell using 85-nm Si (red line) was less than that of the cell using 100-nm Si (black line), demonstrating that less Li$^+$ accumulation occurs in 85-nm Si than in 100-nm Si. Furthermore, the cell using 85-nm Si has better cyclability than that using 100-nm Si (Fig. 3h).

Figure 4a shows the pressure evolution on graphite, calculated from the shift in the XRD peak (stage 1), in full cells with silicon–graphite anodes using high pellet density (HPD)-graphite (denoted as Cell C, black circles) and low pellet density (LPD)-graphite (denoted as Cell D, red circles). Here, LPD-graphite is harder than HPD-graphite; the harder graphite improves the cyclability (see Supplementary Methods 1 and Supplementary Fig. 2 for details). The corresponding voltage profiles of Cell C (black line) and Cell D (red line) over a cycle are also shown in Fig. 4a. The pressure on graphite increases during full-cell charging.

Figure 4b shows the amounts of Li$^+$ in stage 1 and stage 2 for when using HPD-graphite and LPD-graphite. During full-cell charging and subsequent discharging, the maximum amount of Li$^+$ in stage 2 for HPD-graphite is larger than that for LPD-graphite. This observation indicates that the pressure-induced transition from stage 2 to stage 1 in HPD-graphite is energetically more difficult than that in LPD-graphite. The lithiation rate in HPD-graphite decreases above the critical pressure of HPD-graphite (~0.33 GPa), as shown in Fig. 4a and b.

Figure 4c shows the cycling performance of prismatic cells using LPD-graphite and 85-nm Si (red circles) and HPD-graphite and 85-nm Si (black circles). The prismatic cell using LPD-graphite exhibits much better performance, with more than 80% capacity retention after 750 cycles (the Coulombic efficiency of this cell is shown in Fig. 4d), whereas the capacity of the cell using HPD-graphite decreases sharply after 200 cycles (the DC–IR charactieristics of these cells are given in Supplementary Fig. 20). The standard capacity and volumetric energy density of these

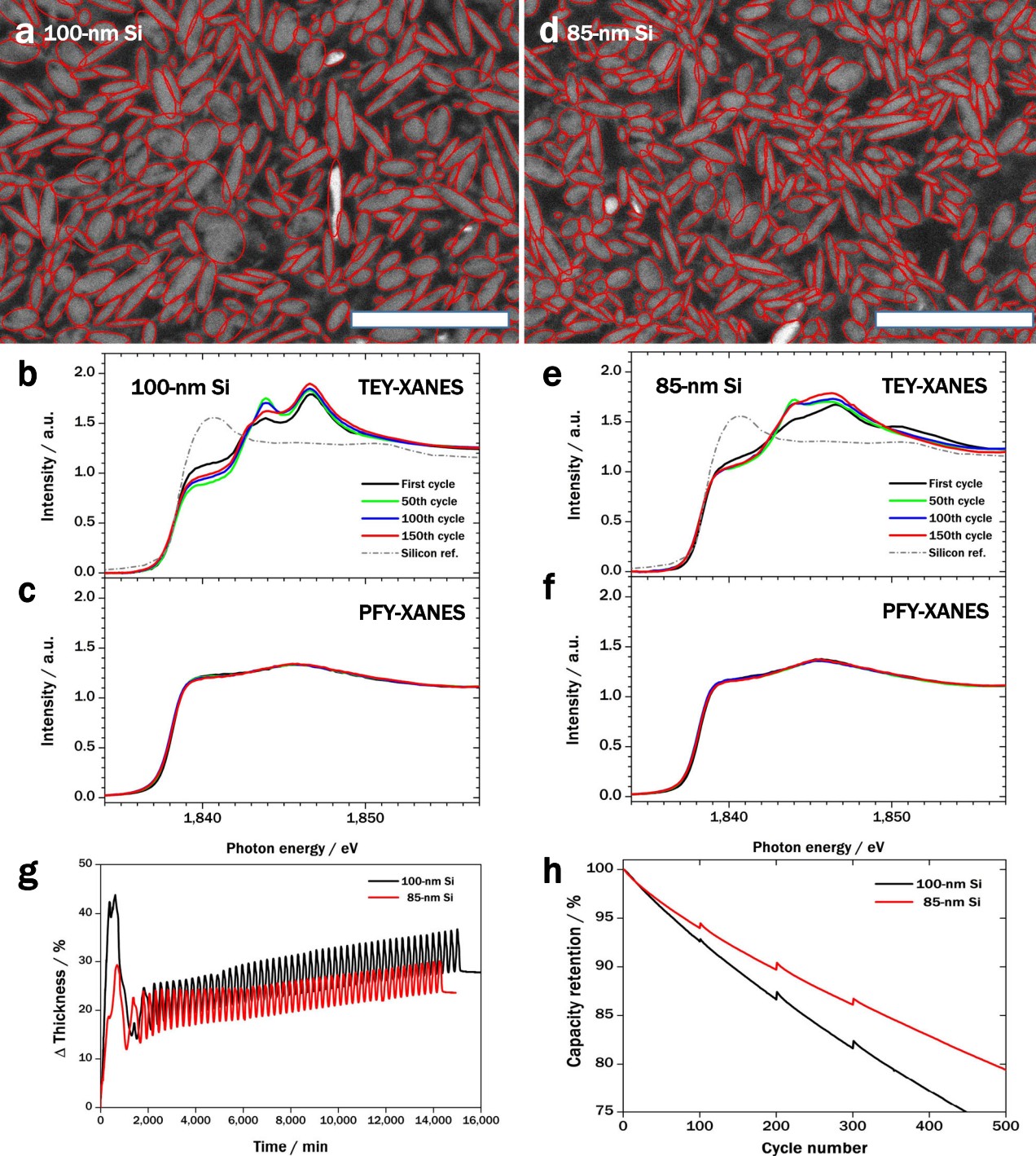

**Fig. 3 Effect of particle size on silicon degradation. a, d** Scanning electron microscopy (SEM) micrographs of SSC cross-sections containing 100- and 85-nm Si particles, respectively (scale bars: 500 nm). The boundaries of the silicon particles are marked in red. **b, e** Total-electron-yield (TEY)-X-ray absorption near edge structure (XANES) spectra of cells containing 100- and **e** 85-nm Si particles, respectively, showing surface degradation of silicon particles with cycling. **c, f** Partial-fluorescence-yield (PFY)-XANES spectra of cells containing 100- and 85-nm Si particles, respectively, showing little change in the bulk properties of silicon particles after cycling. **g** Increase in thickness of cells containing 85- and 100-nm Si particles (black and red lines, respectively). **h** Performance of cells containing 85- and 100-nm Si particles (black and red lines, respectively).

cells were 8.7 Ah and 665 Wh L$^{-1}$, respectively. A simple volumetric calculation indicates that this energy density would correspond to 800 Wh L$^{-1}$ for a prismatic cell with a capacity of 87 Ah, considering the reduction in dead volume in larger cells for practical EV batteries. Thus, graphite capacity fading due to silicon expansion in long-term cycle can be relieved by increasing the graphite hardness.

Numerous ways to engineer the design of the silicon/graphite composite anodes to decrease the internal redox reaction can be investigated in future work. For example, changing the design of the anode to increase the accommodation sites for Li$^+$ in silicon and prevent over-lithiation can decrease the internal redox reaction during CV charging, thus enhancing the cell performance (see Supplementary Notes 16 and 17 for details).

**Table 1 Size of silicon particles in SSC.**

|  | Average length of long axis (nm) | Average length of short axis (nm) | Equivalent radius (nm) |
|---|---|---|---|
| 100-nm Si | 99.91 | 37.76 | 27.61 |
| 85-nm Si | 85.6 | 33.25 | 24.21 |

Average dimensions of silicon particles obtained from SEM micrographs.

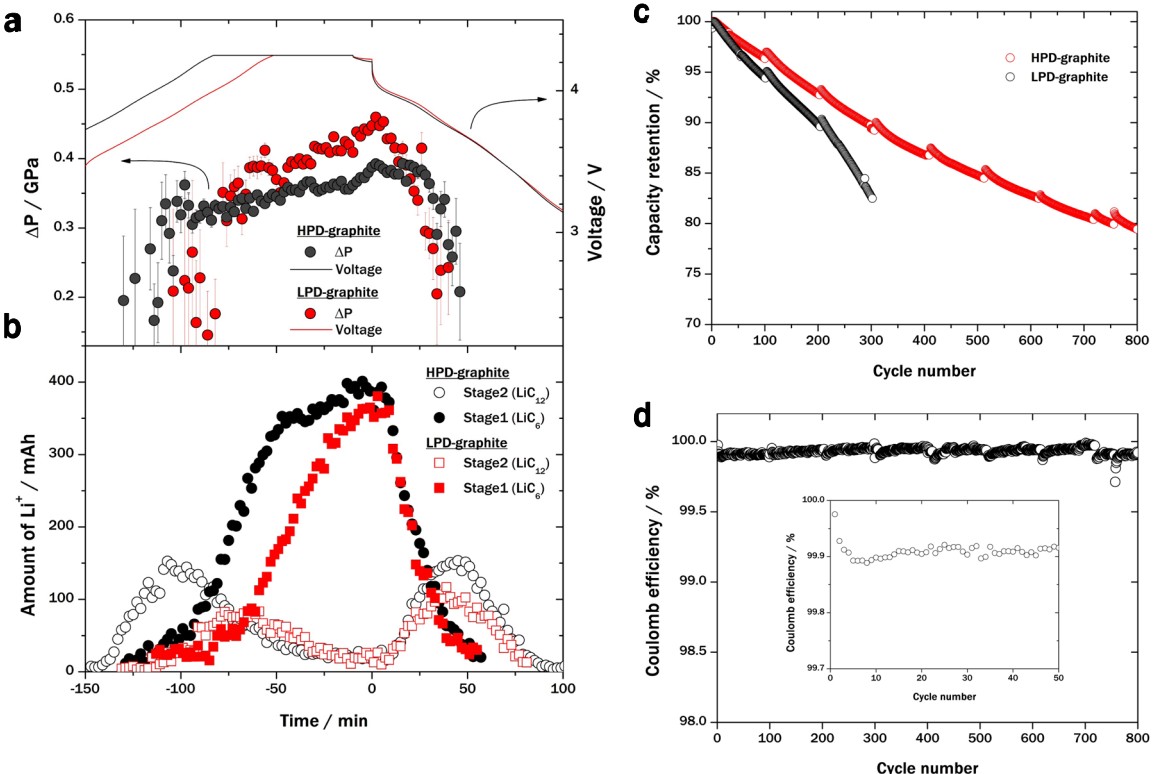

**Fig. 4 Capacity degradation of graphite due to volume expansion of Li$_x$Si. a** Evolution of pressure on graphite ($\Delta P$; circles) and voltage (lines) of Cells C and D, respectively, during cycling. The pressure develops owing to the expansion of Li$_x$Si and subsequent Li$^+$ intercalation into graphite. The discharge start time is set to 0 min. **b** Amounts of Li$^+$ in graphite during stage 1 (filled) and stage 2 (open) in Cells C and D, respectively. **c** Dependence of cycling performance of prismatic cells with a capacity of 8.7 Ah on the physical properties of graphite. Black circle: High pellet density (HPD)-graphite; red circle: low pellet density (LPD)-graphite. **d** Coulombic efficiency of prismatic cell using LPD-graphite. Inset: magnified view of the first 50 cycles. The errors in evolution of pressure on graphite (**a**) are quite large owing to small intensities of stage 1 peaks when the stage 1 partition in graphite is small.

## Discussion

In a blended electrode, Li$^+$ crosstalk is possible between the electrode components if the component materials have different working potentials. However, there are limited analytical methods to identify the Li$^+$ content in Li–Si alloys, which has thus far prevented the observation of Li$^+$ crosstalk in silicon–graphite anodes of full cells. In this study, the graphite capacity was calculated from XRD results, and the silicon capacity was obtained by subtracting the graphite capacity from the full capacity. The internal redox reaction between silicon and graphite in a silicon–graphite anode, as well as the subsequent interplay between the electrochemical reaction and mechanical stress, has been revealed by comprehensive mechano-electrochemical studies in a complete, full-cell-scale long-term cycling analysis in conjunction with analyses of the volume and pressure evolution.

The reversal of chemical potentials between surface-localised Li$_x$Si and Li$_y$C$_6$ drives the mass transport of Li$^+$ from silicon to graphite during CV charging. This internal redox reaction induces a small amount of Li$^+$ to be transported to the interior of the silicon particles at the end of the charging stage owing to

depressurisation of the inner core upon delithiation of the shell. Subsequently, the energy barrier and inhomogeneity of the electrochemical reaction in the silicon particle cause Li$^+$ to remain inside the silicon particle during discharging (Fig. 5a). As a result, Li$^+$ accumulates inside the silicon particles during cycling, causing capacity fading and pressure build-up. The increased pressure on graphite suppresses Li$^+$ intercalation into graphite and depresses the capacity of graphite (Fig. 5b).

By considering these cell degradation mechanisms, the cycling performance can be enhanced by decreasing the silicon particle size and increasing the hardness of graphite, which increase the homogeneity of the electrochemical reactions in silicon and graphite, respectively. Moreover, the decrease in silicon utilisation can reduce the internal redox reaction, allowing tailoring of the cycling performance while minimising adverse effects, such as increased thickness and decreased capacity of full cells. In particular, the implementation of a high-capacity prismatic cell with a high volumetric energy density and decent cycle life can pave the way for the future commercialisation of silicon–graphite anodes.

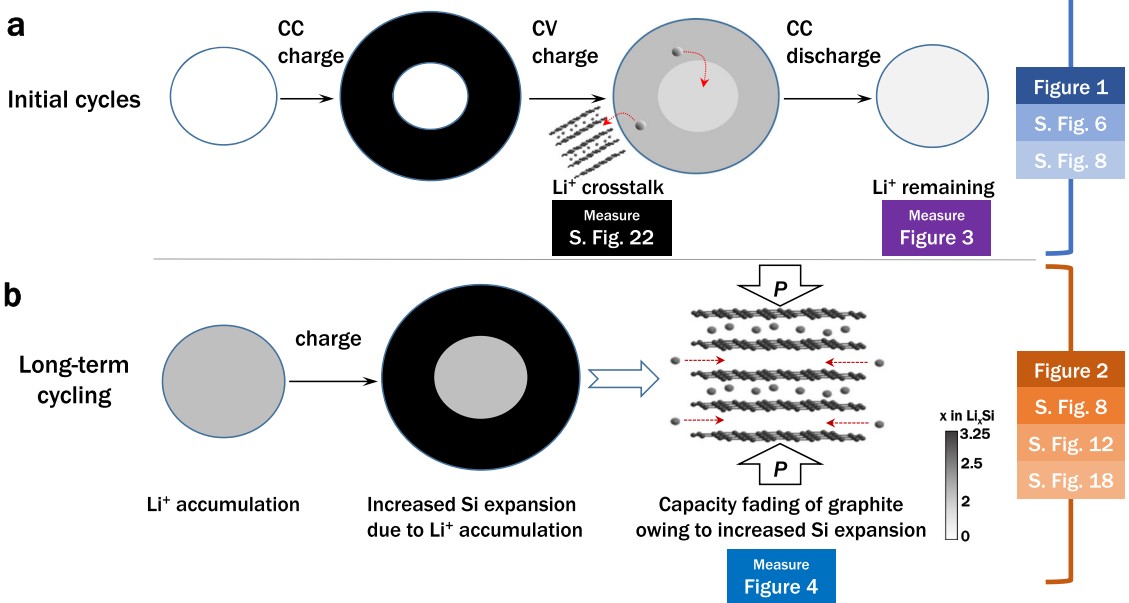

**Fig. 5 Schematic of degradation mechanism. a** In the initial cycles, two-phase lithiation of silicon particles occurs during constant current (CC) charging owing to hydrostatic stress in the particle, and $Li^+$ crosstalk between silicon and graphite occurs during constant voltage (CV) charging. When crosstalk occurs, $Li^+$ penetrates into the silicon core and remains after discharging. **b** During long-term cycling, the accumulation of $Li^+$ in the silicon core increases silicon expansion, which intensifies the pressure (*P*) on graphite; the capacity of graphite is therefore depressed by the pressure-induced staging transition. Figures corresponding to Fig. 5a and **b** are labelled on the right side (S. Fig. denotes a Supplementary Figure). To relieve the long-term detrimental effects, we modified the materials and electrode design. Figures corresponding to improvements to the materials and electrode design are labelled below the corresponding detrimental effects.

## Methods

**Preparation of cells**. Pouch cells with four different electrode chemistries (denoted as Cells A, B, C, and D) were prepared for *operando* XRD experiments. The pouch cells comprised two double-side-coated cathodes (dimensions of 33.4 × 87 mm), one double-side-coated anode, and two single-side-coated anodes (dimensions of 34.4 × 88.5 mm), separated by a ceramic-coated polyethylene porous film (ceramic-coated separator, Toray). The cathodes comprised $Li_{1.0}Ni_{0.88}Co_{0.08}Mn_{0.04}O_2$ (NCM) (97 wt%), carbon black (1.5 wt%, Cabot Co.), and polyvinylidene difluoride (PVDF; 1.5 wt%, Solef). The anodes were prepared with areal capacities of 4.4 or 6 mAh cm$^{-2}$; the former comprised 14.9 wt% SSC, 82.1 wt% graphite, and 3 wt% binder (denoted as SSC–HPD-graphite(4.4)); the latter comprised 16.5 wt% SSC, 80.5 wt%, and 3 wt% binder, with either a high or low pellet density (denoted as SSC–HPD-graphite(6) and SSC–LPD-graphite(6), respectively). The SSC composite was prepared by coating 100-nm Si particles with carbon by chemical vapour deposition and carbonization (see Supplementary Methods 1 for details and Supplementary Fig. 1 for the electrochemical properties of SSC). A graphite-only anode with an areal capacity of 4.4 mAh cm$^{-2}$ comprising 96 wt% HPD-graphite and 4 wt% binder was also prepared (denoted as HPD-graphite(4.4)). The anodes in Cells A, B, C, and D were HPD-graphite(4.4), SSC–HPD-graphite(4.4) (see Supplementary Methods 1 for details), SSC–HPD-graphite(6), and SSC–LPD-graphite (6). The negative to positive (*n/p*) ratios of Cells A, B, C, and D were 1.08, 1.03, 1.03, and 1.03, respectively, and the capacities were 480, 430, 630, and 630 mAh, respectively. All the cells were filled with 1.15 M lithium hexafluorophosphate (LiPF$_6$, Panax E-Tec) dissolved in a mixed solvent of fluoroethylene carbonate (FEC, Panax E-Tec), ethylene carbonate (EC, Panax E-Tec), ethylmethyl carbonate (EMC, Panax E-Tec), and dimethyl carbonate (DMC, Panax E-Tec) in a volume ratio of 7:7:46:40 (FEC/EC/EMC/DMC) as electrolyte.

To investigate the effects of silicon particle size on volume expansion and cyclability, anodes were prepared using 100- and 85-nm Si particles embedded in the SSC. The anodes comprised SSC (14.9 wt%), graphite (82.1 wt%), and binder (3 wt%; modified lithium polyacrylate/polyvinyl alcohol, $M_w$ = 500,000, Sumitomo). The specific capacities of the SSCs with 85- and 100-nm Si particles were identical.

For the in situ thickness measurements, pouch cells were fabricated using a double-side-coated cathode (dimensions of 32 × 40 mm) and a single-side-coated anode (dimensions of 35 × 85 mm), separated by ceramic-coated separator film. The stack was folded in half lengthways to create a pouch structure, with the cathode on the inside. Any gases produced during cycling are therefore expelled from the stack into the designed space inside the pouch cell. The cathode powder comprised $Li_{1.0}Ni_{0.6}Co_{0.2}Mn_{0.2}O_2$ (80 wt%) and $Li_{1.0}Ni_{0.8}Co_{0.1}Al_{0.1}O_2$ (20 wt%); the anodes were the same as those for Cells A–D. These cells were then filled with 1.15 M LiPF$_6$ dissolved in EC, EMC, and DMC in a volume ratio of 2:4:4 to minimise gas evolution during the formation process. The areal capacity and *n/p*

ratio of these cells were 3.4 mAh cm$^{-2}$ and 1.03, respectively. For the cell with the graphite-only anode, the areal capacity and *n/p* ratio were 3.4 mAh cm$^{-2}$ and 1.08, respectively.

To measure the compression force, similar pouch cells but with a higher capacity of 1.5 Ah were fabricated to increase the resolution of the force measurements. Cells were prepared with silicon–graphite and graphite-only anodes, wherein the electrodes were the same as those in Cells C and A, respectively.

The cycling performance of the anode materials was tested using mini-18650 cells, which are essentially cylindrical-type 18650 cells with a hollowed-out cylindrical polypropylene insert that reduces the effective internal cell volume and the total capacity of the cell. The mini-18650 cells comprised one double-side-coated cathode (dimensions of 54 × 200 mm) and one double-side-coated anode (dimensions of 57 × 240 mm), separated by ceramic-coated separator film. The cathode powder comprised $Li_{1.0}Ni_{0.6}Co_{0.2}Mn_{0.2}O_2$ (80 wt%) and $Li_{1.0}Ni_{0.8}Co_{0.1}Al_{0.1}O_2$ (20 wt%). The cells were filled with 1.15 M LiPF$_6$ dissolved in FEC, EC, EMC, and DMC in a volume ratio of 7:7:46:40 as electrolyte. The areal capacity and *n/p* ratio of these cells were 3.4 mAh cm$^{-2}$ and 1.03, respectively.

A prismatic cell comprising 23 double-side-coated cathodes and 24 double-side-coated anodes separated by ceramic-coated separator films was fabricated in a can with internal dimensions of 10 × 80 × 60 mm. The electrodes in this prismatic cell were almost the same as those in Cell D, but the silicon particles embedded in the SSC of the anode were 85-nm Si. The standard capacity, gravimetric energy density, and volumetric energy density of this cell were 8.7 Ah, 318 Wh kg$^{-1}$, and 665 Wh L$^{-1}$, respectively. In a prismatic cell with a capacity of 87 Ah, this energy density corresponds to 800 Wh L$^{-1}$.

All cell fabrication processes, including electrode coating and assembly of pouch cells, were conducted in a dry room with a dew point below −40 °C. The specifications of the cells are tabulated in Supplementary Table 2. Refer to Supplementary Methods 1 for the electrode materials and preparation details.

**Electrochemical evaluation**. All pouch cells were placed between two aluminium plates under an initial pressure of 35 Pa and then connected to a battery cycle testing system (TOSCAT, Toyo System Ltd.). After the formation stage and two standard cycles, the cells were cycled at 25 °C, with a 10 min rest period between cycles. For each cycle, the full cells with areal capacities of 4.4 mAh cm$^{-2}$ (Cells A and B) and 6 mAh cm$^{-2}$ (Cells C and D) were charged to 4.3 and 4.25 V, respectively, at C-rates of 1 C and 0.5 C, respectively, and held in CV charging mode until the charging current reduced to below 0.05 C. After a 10 min rest period, the cells were discharged to 2.8 V at a C-rate of 1 C (Cells A and B) or 0.5 C (Cells C and D). The distance between the two aluminium plates was kept constant

during cycling. The large-cell performance was characterised using the prismatic cell under identical test conditions to those for Cell D.

A three-electrode pouch cell was fabricated to characterise the individual electrochemical behaviour of the cathode and anode in the full cells. Lithium foil was placed in the centre of the bottom side of the pouch to minimise the overpotential[54]. During cycling, the individual electrode potentials were recorded against a lithium reference electrode.

For the mini-18650 cells, the formation and standard schemes were 0.1 C and 0.2 C CC–CV charging and CC discharging modes, respectively, between 4.2 and 2.8 V with a cut-off of 0.05 C at 25 °C. The cells were then cycled using 1 C CC–CV charging and CC discharging modes between 4.2 and 2.8 V at 25 °C.

During the cycling tests of the full cells, the DC-IR was measured every 100 cycles by charging the cell to 50% SOC at a C-rate of 0.2 C. After the measurement, the cells were charged to the upper cut-off voltage at a C-rate of 0.2 C, and the cycling tests were resumed.

**Operando XRD**. The high X-ray intensity required for the *operando* measurements was achieved with the XRD installed at BL15XU in SPring-8 (Hyogo, Japan). All pouch full cells were fully discharged to 2.8 V at a C-rate of 0.2 C and then placed between two aluminium plates containing a beryllium window, in the same manner as for the electrochemical evaluation. The photon energy and wavelength transmitted through the pouch cell were 18.987 keV and 0.65297 Å, respectively. All diffraction profiles were collected using four detectors (1D multichannel) in the $2\theta$ range of 0–50°. Sequential XRD profiles and corresponding electrochemical data were obtained by exposure for 10 s every 1 min at operational rates of 1 C and 2 C, and every 2 min at an operational rate of 0.5 C.

**Calculation of lithium-ion contents in SSC and graphite**. To calculate the Li+ contents in the SSC and graphite components, lithium half cells with SSC, graphite, or an SSC–graphite composite as the working electrode were prepared. The areal capacity of each electrode was 1.5 mAh cm$^{-2}$. The charge/discharge potentials were measured at 0.05 C (CV at 0.02 C), which was presumed to be slow enough to measure the pseudo-equilibrium potential. Considering the mass ratio of SSC and graphite in the composite electrode, the Li+ content in each component at a certain potential was calculated by using the corresponding SOC *vs.* potential profiles for the half cells with SSC and graphite working electrodes. The calculated SOC *vs.* potential profile for the composite electrode was well-matched with the measured profile. Through this process, the individual SOCs of each material were deconvoluted and simulated for half cells with various compositions.

**Force measurements**. To measure the pressure on the graphite during cycling, in situ force measurements were conducted. A pouch cell with a capacity of 1.5 Ah was designed with a vacant space inside, such that any gas evolved during cycling would be expelled into this space without changing the cell thickness. The electrodes in this pouch cell were almost the same as those in Cell C. For comparison, we also prepared a pouch cell with a graphite-only anode, which had the same electrodes as those in Cell A.

The force on the pouch cell was measured during cycling tests[55,56]. The constraint fixture consisted of the pouch cell in series with a load cell. The pouch cell and load cell were fastened together between two stainless steel fastening plates with nuts and bolts. A force-distributing plate was also placed between the pouch cell and load cell; therefore, the change in cell thickness would also change the force exerted on the force-distributing plate. This force was measured with a load cell (OBUH-50; Bongshin Loadcell Co. Ltd.) with a capacity of 50 kgf connected to strain gauge panel metres with a data logger (GL240, Omni Instruments).

The initial force was adjusted by tightening the nuts with a torque-controlled wrench. For in situ measurements, the pouch cells were connected to the TOSCAT battery cycle testing system. The cells were cycled under 0.5 C CC–CV charging and 0.5 C CC discharging modes between 4.25 and 2.8 V at 25 °C.

**Scanning electron microscopy (SEM)**. Anode cross-sections were prepared using an ion-beam cross-section polisher combined with a nitrogen cooling system and an air isolation system (IB-19520CCP, JEOL) to prevent ion-beam damage and exposure to ambient air. These specimens were then transferred to the SEM chamber (Helios NanoLab 450HP, FEI) using an air isolation transfer system. The silicon nanoparticles embedded in carbon in the SSC were imaged in secondary electron mode at an accelerating voltage of 2 kV and magnification of ×250,000.

**X-ray absorption fine structure (XAFS) spectroscopy**. Ex situ Si K-edge XAFS spectra were measured at BL-10 in the Ritsumeikan SR Center. After fully discharging the cells, they were disassembled in an Ar-filled glovebox. The anodes were rinsed in DMC for 5 min, set on carbon-taped specimen holders, loaded in an airtight vessel, and then transferred to the BL-10 chamber without exposure to ambient air. The vessel was immediately evacuated, and the specimens were loaded into the measurement chamber at a vacuum pressure of $5 \times 10^{-8}$ Pa. The photon beam energy was in the range of 1,000–2,500 eV with a resolution of 0.5 eV or less. The Si K-edge XANES spectra of the anode were recorded in TEY (probing depth ~1 nm) and PFY (probing depth ~10–10$^2$ nm) modes[57,58].

**Thickness measurements**. In situ thickness measurements of single-stack pouch cells were conducted during cycling using a custom load-cell tester. After placing a 2 kg stainless steel block on the cells, they were connected to the TOSCAT battery cycle testing system. The formation and standard schemes of the cells were 0.1 C and 0.2 C CC–CV charging and discharging modes, respectively, between 4.2 and 2.8 V with a cut-off of 0.05 C at 25 °C. The cells were then cycled using 0.5 C CC–CV charging and discharging modes between 4.2 and 2.8 V at 25 °C. The thickness of each cell was monitored during cycling using the gap sensor of a non-contact displacement measuring system (Linear Gauge System, Mitutoyo).

## Data availability

The data that support the findings of this study are available from the corresponding authors upon reasonable request.

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

## Acknowledgements

This work was supported by funds from Samsung Electronics Co. Ltd. A.C.L and J.G.S. acknowledge support from the National Research Foundation of Korea (NRF) grant funded by the Korea government (MSIT) (No. 2019R1A2C1090304).

## Author contributions

Y.-G.R. and I.T.H. proposed and supervised the research; J.M., I.T.H., and Y.-G.R. designed the study; J.M., S.W., and H.J. performed the electrochemical analysis; S.C. conducted the in situ thickness measurements; Sh.L. performed the image analysis; K.I. and Y.K. carried out the operando XRD and XAFS measurements; J.M. analysed the XRD and XAFS data; A.U. and B.C. conducted the in situ force measurements; J.M., A.C.L, J.G.S., and J.Yoo carried out in situ TEM; Sy.L. and Y.H. performed ICP measurements; W.B. conducted EPMA experiments; and J.Yoon and J.L. performed the electrochemical analysis of the prismatic cells. J.M., H.C.L., and Y.-G.R. interpreted the data and wrote the manuscript with input from all authors.

## Competing interests

The authors declare no competing interests.
