## [Peer Review File · Nature Communications]

Reviewer #1 (Remarks to the Author):

The manuscript is improved by previous revisions. I recommend it to be published in Nature Communications after major revision. First, I suggest authors to adjust the manuscript to be more reader friendly. Second, is there any other visible characterization to further prove the crosstalk between silicon and graphite?

Reviewer #2 (Remarks to the Author):

There is large amount of data included in this manuscript. The cycling performance of the large-format full cell impressive considering the high areal capacities used in the full cell demonstration. The superior cycling stability should be majorly attributed to the SSC silicon-carbon composite developed in this project, rather than the particle sizes of Si and the hardness of graphite used in the Si-graphite composite electrode. However, this manuscript barely discussed the electrochemical properties of the SSC. The author mentioned that the long-term capacity degradation is due to the lithium trapping in the Si and degradation of graphite, which have been well known for such composite electrodes. However, the authors said that the lithium trapping is due to Li crosstalk, which cannot be sufficiently supported by the data. In addition, the degradation of graphite component was attributed to the accumulation of Li ions inside of Si particles, which generates the pressure to limit the lithiation of graphite. This statement is not accurate, since the amount of the trapped Li is much less than that for the lithiation process. The graphite component works fine under such large pressure caused by lithiation in Si component. Please see details below.

The statement: "Li⁺ remaining in silicon due to Li⁺ crosstalk between silicon and graphite" requires more evidence. It is true that there is Li⁺ crosstalk between Si and Graphite, which has been reported previously. The lithium ion diffusion within the Si particles is not the limiting factor, compare to the sluggish lithium ion diffusion at the interface between Si and electrolyte (reference: Nano Lett. 2016, 16, 12, 7394–7401). There always has some lithium ion left behind in the Si particle, but probably at the surface area. The crosstalk may enhance the lithiation of the Si core area, which may increase the utilization of Si. Without the Li cross talk, the lithium ions can also be trapped inside of Si particles.

There is no SEI analysis for the SSC anode. It is not clear that how much capacity loss is due to the SEI formation for the SSC anode. The authors stated that the accumulation of the remaining Li causes the volume expansion of Si particles, which causes the large pressure for the graphite component, leading to the capacity degradation of graphite. But, as compared to the large volume expansion during the Si lithiation, the remaining of Li ions only cause less volume expansion. However, the authors claimed that the pressure caused by the accumulation of Li ions becomes a major player in generating the pressure for the graphite component.

Responses for Reviewers' Comments

Responses for Reviewer #1's comments

The manuscript is improved by previous revisions. I recommend it to be published in Nature Communications after major revision. First, I suggest authors to adjust the manuscript to be more reader friendliness. Second, is there any other visible characterization to further prove the crosstalk between silicon and graphite?

Author reply: We gratefully appreciate the reviewer #1's thoughtful comments so far.

We thoroughly checked again the previous studies on silicon-graphite anodes and analytic techniques for unveiling their mechanisms. Thus, we revised introduction in our manuscript for helping readers understanding our main issue, goal, and strategies and the whole manuscript was revised and edited by a professional editing service. We believe that this revision enhances reader friendliness. Newly added contents in Introduction are as follows.

Manuscript lines 61-63 & 65-70

It is necessary to understand the detailed interactions between silicon and graphite in silicon-graphite composite anodes, such as the (de)lithiation kinetics, electrochemical behaviour, mechanical response, and their interplay with cycling stability²².

Up to now, despite efforts to produce reliable analytical techniques that can identify the Li⁺ content in amorphous Li-Si alloys²³⁻²⁷, amorphous lithium silicide does not produce diffraction peaks, which has hindered investigations of the interaction between silicon and graphite during full-cell operation; its visualisation has also been challenging because of the complexity of the experimental environment²⁸.

References

22. Chae, S., Choi, S.-H., Kim, N., Sung, J. & Cho, J. Integration of graphite and silicon anodes for the commercialization of high-energy lithium-ion batteries. *Angew. Chem. Int. Ed.* **59**, 110–135 (2020).

28. Boebinger, M. G., Lewis, J. A., Sandoval, S. E. & McDowell, M. T. Understanding transformations in battery materials using in situ and operando experiments: progress and outlook. *ACS Energy Lett.* **5**, 335–345 (2020).

We newly added STEM-EELS images of the chemical distribution in SSC to visualize Li⁺ accumulation in silicon. After 100 cycles, silicon particles are in a state of lithiation as shown in Supplementary Fig. 12, whereas silicon particles have little lithium after the first cycle (Supplementary Fig. 11). These visualizations provide us a proof of Li⁺ crosstalk. We added lines 217-219 in manuscript and Supplementary Note 9 about TEM measurements as follows.

Manuscript line 217-219

This Li⁺ accumulation was also visualised visualized through scanning transmission electron microscopy-electron energy loss spectroscopy (STEM-EELS) (see Supplementary Note 9 and Supplementary Figs. 11 and 12).

SI lines 382-412 & Supplementary Figs. 11 and 12

Supplementary Note 9. Transmission electron microscopy (TEM)

Ex situ transmission electron microscopy (TEM) analyses were carried out using a JEM-

ARM200F-G microscope (JEOL) with electron energy loss spectroscopy (EELS; Gatan Enfium) and energy dispersive spectroscopy (EDS; SDD type) detectors. The cell was disassembled in an Ar-purged glovebox, and the electrode was rinsed in DMC for 5 min and then dried under vacuum for 30 min. To avoid sample contamination and reaction upon air exposure, a vacuum transfer TEM holder (Gatan Model 648) and custom transfer vessel for FIB were used. All samples were moved from the FIB transfer vessel to the vacuum transfer TEM holder in an Ar-purged glovebox.

In scanning transmission electron microscopy (STEM) mode, the specimen is illuminated only at the position where the focused electron beam is transmitted, so EELS and EDS signals generated from electron excitation have high spatial accuracy. We investigated the chemical distribution of the SSC and SEI on SSC after the first and 100th cycles. The SEI, consisting mainly of LiF, grows during the formation stage on the coal tar pitch surrounding the SSC particles (see Supplementary Methods 1), as shown in Supplementary Fig. 11²⁷. After 100 cycles, the Li-K edge EELS image exhibits bright and dark red areas, as shown in Supplementary Fig. 12b. The bright part surrounds the pitch, and consists mainly of LiF (Supplementary Fig. 12d), similarly to that on the SSC particles after the first cycle (Supplementary Fig. 11e). The dark part is lithium silicide (Supplementary Fig. 12d). Therefore, Li⁺ had accumulated inside the silicon particles by the 100th cycle. These results agree well with the results in Supplementary Note 8, which illustrates the accumulation of Li⁺ in silicon during long-term cycling.

Along with PFY-XANES result (Figs. 3c and f), this STEM-EELS results exhibit that LiF as a component of SEI forms mainly on the carbon components of SSC. The ratio of graphite and the carbon in SSC is 82.1 : 7.1 (= 14.9 wt% (wt% of SSC in anode) × 47.7% (carbon ratio in SSC)). During Cell B (capacity of 430 mAh) operation, the lithium loss by SEI at first and 250th cycles is 74.9 and 81.31 mAh, respectively (Supplementary Table 1). Then, the roughly estimated lithium losses by SEI on SSC at first and 250th cycles is 6.0 and 6.5 mAh, respectively. The specific properties of SEIs on different components as well as the ratios of volumes and surfaces of materials could affect the calculation result, but might not significantly change the estimation.

Supplementary Figure 11. Formation of solid electrolyte interphase (SEI) on surface-treated

silicon/carbon (SSC) in Cell B after discharge of first cycle. a ADF image of SSC and b, c corresponding energy dispersive spectroscopy (EDS) mapping images for b Si and c F. Scale bars: 0.5 μm . d Corresponding electron energy loss spectroscopy (EELS) mapping of Li K-edge of SSC and e EELS spectra at points 1 and 2 on the EELS mapping image.

Supplementary Figure 12. Solid electrolyte interphase (SEI) on surface-treated silicon/carbon (SSC) and Li^+ accumulation in SSC of Cell B after discharge of 100th cycle. a ADF image and b, c corresponding electron energy loss spectroscopy (EELS) mappings of b Li K- and c Si L-edge of SSC. Scale bars: 2 μm . d Li K-edge EELS spectra at points 1 and 2 on the Li K-edge mapping image.

Supplementary reference

27. Huang, W. et al. Dynamic structure and chemistry of the silicon solid-electrolyte interphase visualized by cryogenic electron microscopy. *Matter* **1**, 1232–1245 (2019).

Responses for Reviewer #2's comments

There is large amount of data included in this manuscript. The cycling performance of the large-format full cell impressive considering the high areal capacities used in the full cell demonstration.

Author reply: We deeply appreciate the reviewer #2's insightful review as well as comment on the significance of our work. We revised abstract for focusing the cycling performance of the large-format full cell as follows.

Manuscript lines 32-34

The resultant full cell with an areal capacity of 6 mAh cm⁻² had a cycle life of >750 cycles the volumetric energy density of 800 Wh L⁻¹ in a commercial cell format.

The superior cycling stability should be majorly attributed to the SSC silicon-carbon composite developed in this project, rather than the particle sizes of Si and the hardness of graphite used in the Si-graphite composite electrode. However, this manuscript barely discussed the electrochemical properties of the SSC.

Author reply: We are grateful for reviewer #2's comment about a lack of the electrochemical performance of SSC. We added the specific capacity retention and potential profile at the first cycle after formation in the half cell of SSC as shown in Supplementary Fig. 1. We also refer to this content in manuscript lines 262-264.

We agree with the reviewer #2's opinion, cycling stability is attributed to the SSC. Basically, SSC delays penetration of electrolyte into silicon particles, which reduces the SEI formation over cycling and makes the Coulombic efficiency of full cell stable. However, we should have employed silicon-graphite anode for the resolution of excessive Li⁺ loss during formation due to low initial Coulombic efficiency of SSC. Consequently, graphite as well as graphite-silicon compatibility affect the cycle performance. Supplementary Fig. 2 exhibits explicitly the cycle performance dependence on the hardness of graphite, in which, all graphites we used are the commercial materials for commercial batteries. The cycle performance dependence on the silicon size was also exhibited in Figure 3h.

Manuscript lines 262-264

... (initial Coulombic efficiency: 86.4%, the capacity retention at 200th cycle: 86.6%, see Supplementary Fig. 1 and Supplementary Note 9 for electrochemical properties and SEI formation on SSC, respectively).

Supplementary Fig. 1

Supplementary Figure 1. Electrochemical properties of surface-treated silicon/carbon (SSC) composite. a Specific capacity retention of SSC at a C-rate of 0.5C. **b** Voltage profile of SSC during the first cycle.

The author mentioned that the long-term capacity degradation is due to the lithium trapping in the Si and degradation of graphite, which have been well known for such composite electrodes. However, the authors said that the lithium trapping is due to Li crosstalk, which cannot be sufficiently supported by the data.

Author reply: We are very grateful for reviewer #2's constructive comment about the origin of lithium trapping in silicon.

We are aware of lithium trapping issue, e.g., the recent studies of L. Nyholm group (*Energy Environ. Sci.* **10**, 1350-1357 (2017)) and Y. Cui group (*Sci. Adv.* **5**, eaax0651 (2019)). The former claims the effect of Li⁺ trapping in silicon accounts for ca. 30% of the initial Li loss for the first cycle; the latter claims that reducing the energy barrier of Li⁺ diffusion minimizes Li⁺ trapping.

In these half-cell studies, typical lithiation scheme is constant current-constant voltage lithiation to 10 mV vs. Li/Li⁺. To our best knowledge, silicon is lithiated fully when the potential drop below 60 mV (McDowell, M. T. *et al. Adv. Mater.* **25**, 4966-4985 (2013)). At the end of delithiation, Li⁺ in the center of silicon traps in the core of silicon explained by two-way diffusion model. In our full-cell system, the DOD of silicon is 80% as shown in Fig. 1c. For this reason, Li⁺ could not insert into the core of silicon if without Li⁺ crosstalk between silicon and graphite. We observed Li⁺ crosstalk through *operando* XRD as well as *in situ* thickness and force measurements. The possibility of Li⁺ crosstalk was explained by thermodynamic calculation in Supplementary Note 4. Moreover, in the silicon-graphite anode, once Li⁺ crosstalk occurs, Li⁺ in the shell can insert the depressurized core of the silicon particle, which was explained by calculation in Supplementary Note 7 (thermodynamic calculation).

For the same reason as the half-cell studies, Li⁺ in the silicon core is remaining at the end of discharge step. As shown in Fig. 2a, OCV decrease after the 250th cycle supports the accumulation of Li⁺ in silicon; Fig. 2b exhibits that the delay region for Li⁺ transfer (indicated by the black box in Fig. 2b) could suggest that Li⁺ is slowly inserted into the silicon particles owing to the presence of accumulated Li⁺ in the inner core of the silicon particles. Consequently, Li⁺ crosstalk facilitates Li⁺ accumulation in the silicon core.

We obtained the additional proofs of Li⁺ accumulation over cycling through various analyses such as *in situ* thickness measurement (Fig. 3g), ICP-AES (Supplementary Fig. 8), and STEM-EELS

(Supplementary Fig. 12). Furthermore, we showed that Li⁺ crosstalk reaction could be regulated in Supplementary Note 16 (Improvement over existing electrode design). We can directly relieve the long-term detrimental effects of Li⁺ crosstalk by changing the design of the anode, maintaining a fixed content of graphite, and increasing the content of SSC to 111% relative to the content of the previous anode. We can enhance the cycling performance. This is a proof that Li⁺ crosstalk is a key factor in Li⁺ accumulation in silicon.

We added lines 217-219 in manuscript and Supplementary Note 9 (STEM-EELS) as follows.

Manuscript lines 217-219

This Li⁺ accumulation was also visualised through scanning transmission electron microscopy-electron energy loss spectroscopy (STEM-EELS) (see Supplementary Note 9 and Supplementary Figs. 11 and 12).

SI lines 382-412

Supplementary Note 9. Transmission electron microscopy (TEM)

Ex situ transmission electron microscopy (TEM) analyses were carried out using a JEM-ARM200F-G microscope (JEOL) with electron energy loss spectroscopy (EELS; Gatan Enfium) and energy dispersive spectroscopy (EDS; SDD type) detectors. The cell was disassembled in an Ar-purged glovebox, and the electrode was rinsed in DMC for 5 min and then dried under vacuum for 30 min. To avoid sample contamination and reaction upon air exposure, a vacuum transfer TEM holder (Gatan Model 648) and custom transfer vessel for FIB were used. All samples were moved from the FIB transfer vessel to the vacuum transfer TEM holder in an Ar-purged glovebox.

In scanning transmission electron microscopy (STEM) mode, the specimen is illuminated only at the position where the focused electron beam is transmitted, so EELS and EDS signals generated from electron excitation have high spatial accuracy. We investigated the chemical distribution of the SSC and SEI on SSC after the first and 100th cycles. The SEI, consisting mainly of LiF, grows during the formation stage on the coal tar pitch surrounding the SSC particles (see Supplementary Methods 1), as shown in Supplementary Fig. 11²⁷. After 100 cycles, the Li-K edge EELS image exhibits bright and dark red areas, as shown in Supplementary Fig. 12b. The bright part surrounds the pitch, and consists mainly of LiF (Supplementary Fig. 12d), similarly to that on the SSC particles after the first cycle (Supplementary Fig. 11e). The dark part is lithium silicide (Supplementary Fig. 12d). Therefore, Li⁺ had accumulated inside the silicon particles by the 100th cycle. These results agree well with the results in Supplementary Note 8, which illustrates the accumulation of Li⁺ in silicon during long-term cycling.

Along with PFY-XANES result (Figs. 3c and f), this STEM-EELS results exhibit that LiF as a component of SEI forms mainly on the carbon components of SSC. The ratio of graphite and the carbon in SSC is 82.1 : 7.1 (= 14.9 wt% (wt% of SSC in anode) × 47.7% (carbon ratio in SSC)). During Cell B (capacity of 430 mAh) operation, the lithium loss by SEI at first and 250th cycles is 74.9 and 81.31 mAh, respectively (Supplementary Table 1). Then, the roughly estimated lithium losses by SEI on SSC at first and 250th cycles is 6.0 and 6.5 mAh, respectively. The specific properties of SEIs on different components as well as the ratios of volumes and surfaces of materials could affect the calculation result, but might not significantly change the estimation.

Supplementary Figure 11. Formation of solid electrolyte interphase (SEI) on surface-treated silicon/carbon (SSC) in Cell B after discharge of first cycle. a ADF image of SSC and **b, c** corresponding energy dispersive spectroscopy (EDS) mapping images for **b** Si and **c** F. Scale bars: 0.5 μm. **d** Corresponding electron energy loss spectroscopy (EELS) mapping of Li K-edge of SSC and **e** EELS spectra at points 1 and 2 on the EELS mapping image.

Supplementary Figure 12. Solid electrolyte interphase (SEI) on surface-treated silicon/carbon (SSC) and Li^+ accumulation in SSC of Cell B after discharge of 100th cycle. a ADF image and b, c corresponding electron energy loss spectroscopy (EELS) mappings of b Li K- and c Si L-edge of SSC. Scale bars: 2 μm . d Li K-edge EELS spectra at points 1 and 2 on the Li K-edge mapping image.

Supplementary reference

27. Huang, W. *et al.* Dynamic structure and chemistry of the silicon solid-electrolyte interphase visualized by cryogenic electron microscopy. *Matter* **1**, 1232–1245 (2019).

In addition, the degradation of graphite component was attributed to the accumulation of Li ions inside of Si particles, which generates the pressure to limit the lithiation of graphite. This statement is not accurate, since the amount of the trapped Li is much less than that for the lithiation process. The graphite component works fine under such large pressure caused by lithiation in Si component.

Author reply: We deeply appreciate reviewer #2's inspirational comment about the amount of remaining Li in silicon and pressure evolutions in and over cycles. We agreed that the statement was

not accurate enough, so we totally reconstructed it.

Supplementary Fig. 18 exhibits evolution of pressure on graphite. As like 250th cycle, the abrupt increase of pressure is enough for delay of transition from stage 2 to stage 1 to occur as shown in Figures 2c and d. The onsets of Li^+ in stage 1 of graphite arise at -154.69 mAh (corresponding time is 39 min) and at -109.44 mAh (38 min) for first and 250th cycles, respectively (blue and red arrows in Figs. 2 c and d). At these points, the pressures on the graphite are 0.290 and 0.329 GPa for first and 250th cycles, respectively. At the points of the maximum lithiation in silicon, -83.02 mAh (49 min) for first cycle and -28.96 mAh (53 min) for 250th cycle, the pressures on graphite are 0.295 and 0.388 GPa for first and 250th cycles, respectively, as shown in Fig. 2c and Supplementary Fig. 18. The increase in pressure impacts on the lithiation in graphite owing to pressure-induced staging transition of graphite.

As shown in Figs. 4a and b, the critical pressure of HPD-graphite for the pressure-induced staging transition would be ca. 0.33 GPa. So, during Cell B operation, at early cycles when pressure does not exceed this critical pressure, there is no significant degradation. However, after pressure is built up and starts to exceed the critical pressure, it might starts to degrade.

We added lines 239-247 and line 314-316 in the revised manuscript.

Manuscript lines 239-247

The onsets of Li^+ in stage 1 of graphite arise at -154.69 mAh (corresponding time: 39 min) and at -109.44 mAh (38 min) for the first and 250th cycles, respectively (blue and red arrows in Figs 2 c and d). At these points, the graphite is experiencing 0.290 and 0.329 GPa of pressure in first and 250th cycles, respectively. At the points of maximum lithiation in silicon (-83.02 mAh (49 min) for first cycle and -28.96 mAh (53 min) for 250th cycle), the pressures on graphite increase to 0.295 and 0.388 GPa for first and 250th cycles, respectively, as shown in Fig. 2c and Supplementary Fig. 18. The increase in pressure impacts the lithiation of graphite. The critical pressure of graphite for the pressure-induced staging transition would be ca. 0.33 GPa.

Manuscript lines 314-316

The lithiation rate in HPD-graphite decreases above the critical pressure of HPD-graphite (~0.33 GPa), as shown in Figs. 4a and b.

Please see details below.

The statement:” Li^+ remaining in silicon due to Li^+ crosstalk between silicon and graphite” requires more evidence. It is true that there is Li^+ crosstalk between Si and Graphite, which has been reported previously.

Author reply: We thanks for the reviewer #2’s comment. We thoroughly checked the previous studies on silicon-graphite anodes again. The list of major reviews and important reports is as follows. To our best knowledge, there is no report about the observation of Li^+ crosstalk in the silicon-graphite anodes. We truly would like to know, if any, previously published papers on the experimental proof of the Li^+ crosstalk between Si and graphite so we can make our arguments stronger by referring them in our paper.

Reviews and reports about silicon-graphite composite/blending anodes

- Review about the role of graphite in SOC control of silicon for long cycling stability and the necessity of understanding the detailed interaction between silicon and graphite
Chae, S. *et al.* Integration of graphite and silicon anodes for the commercialization of high-energy lithium-ion batteries. *Angew. Chem. Int. Ed.* **59**, 110–135 (2020).
- Review about the role of carbon in co-utilization of silicon and graphite

- Wu, J. *et al.* The critical role of carbon in marrying silicon and graphite anodes for high-energy lithium-ion batteries. *Carbon Energy* **1**, 57-76 (2019).
- Report on the role of graphite for improving conductivity in silicon-graphite anode
Du, Z., Dunlap, R. A. & Obrovac, M. N. High energy density calendared Si alloy/graphite anodes. *J. Electrochem. Soc.* **161**, A1698-A1705 (2014).
 - Reports on the composite anodes for long cycling stability and the role of carbon/graphite
Kim, S. Y. *et al.* Facile synthesis of carbon-coated silicon/graphite spherical composites for high-performance lithium-ion batteries. *ACS Appl. Mater. Interfaces* **8**, 12109-12117 (2016).
Ko, M. *et al.* Scalable synthesis of silicon-nanolayer-embedded graphite for high-energy lithium-ion batteries. *Nat. Energy* **1**, 16113 (2016).
Kim, N. *et al.* Fast-charging high-energy lithium-ion batteries via implantation of amorphous silicon nanolayer in edge-plane activated graphite anodes. *Nat. Commun.* **8**, 812 (2017).
 - Report on design and physical properties and of the composite anodes
Otero, M. *et al.* Design-considerations regarding silicon/graphite and tin/graphite composite electrodes for lithium-ion batteries. *Sci. Rep.* **8**, 15851 (2018).

Reviews and reports on internal redox couple in the blended cathodes

Heubner, C., Liebmann, T., Schneider, M. & Michaelis, A. Recent insights into the electrochemical behavior of blended lithium insertion cathodes: A review. *Electrochim. Acta* **269**, 745-760 (2018).

Heubner, C., Lämmel, C., Schneider, M. & Michaelis, A. Temperature-induced compositional redistribution in blended insertion electrodes. *J. Power Sources* **344**, 170-175 (2017).

Heubner, C. *et al.* Insights into the buffer effect observed in blended lithium insertion electrodes. *J. Power Sources* **363**, 311-316 (2017).

Klein, A., Axmann, P. & Wohlfahrt-Mehrens, M. Origin of the synergetic effects of $\text{LiFe}_{0.3}\text{Mn}_{0.7}\text{PO}_4$ spinel blends via dynamic in situ x-ray diffraction measurements. *J. Electrochem. Soc.* **163**, A1936-A1940 (2016).

Kobayashi, T., Kobayashi, Y. & Miyashiro, H. Lithium migration between blended cathodes of a lithium-ion battery. *J. Mater. Chem. A* **5**, 8653-8661 (2017).

The lithium ion diffusion within the Si particles is not the limiting factor, compare to the sluggish lithium ion diffusion at the interface between Si and electrolyte (reference: Nano Lett. 2016, 16, 12, 7394–7401).

Author reply: we thank for reviewer #2's comment. Under the well-controlled environment of crystalline phased silicon in the reference, the fast Li^+ diffusion in silicon was measured; but, it has been still reported that the enhancement of Li^+ diffusion needs to minimize Li^+ trapping in the silicon core such as the report of Y. Cui's group (*Sci. Adv.* **5**, eaax0651 (2019)).

There always has some lithium ion left behind in the Si particle, but probably at the surface area. The crosstalk may enhance the lithiation of the Si core area, which may increase the utilization of Si. Without the Li cross talk, the lithium ions can also be trapped inside of Si particles.

Author reply: We thank for reviewer #2's comment.

As shown in Figs. 2a and b, the potential of anode cannot reach below 60 mV; the DOD of silicon is 85% (at -90 mAh of the x-axis in Fig. 2b) at first cycle. In this reason, the core of silicon cannot be used because of two-phase lithiation during CC charging in the full cell. Li^+ crosstalk during CV charging leads to a little amount of Li^+ penetration into the silicon core, which is explained by

thermodynamic calculation in Supplementary Note 7. Once Li^+ remains in the silicon core, it is hard for the trapped Li^+ to delithiate from silicon owing to the two-way diffusion trapping (*Energy Environ. Sci.* **10**, 1350-1357 (2017)). We added lines 175-176 in manuscript and a reference.

Moreover, Figure 2b indicates that the accumulated Li^+ is in the silicon core. If the trapped Li ions would be in the surface of silicon, the lithiation rate in silicon at 250th cycle would not be the same with that at first cycle during the early stage of charging mode in Fig. 2b; but, the lithiation rates during the early stage (blue lines on Fig. 2b) are the same. Moreover, the delay region for Li^+ transfer (indicated by the black box in Fig. 2b) could suggest that Li^+ is slowly inserted into the silicon particles owing to the presence of accumulated Li^+ in the inner core of the silicon particles.

Of course, the lithium can be found in the form of SEI, irreversible form at the surfaces of silicon and SSC as shown in Figs 3b and e as well as Supplementary Figs. 11 and 12.

Manuscript lines 175-176

This trapping of Li^+ in the silicon core is similar to the two-way diffusion trapping model⁴⁸.

Reference

48. Rehnlund, D. et al. Lithium trapping in alloy forming electrodes and current collectors for lithium based batteries. *Energy Environ. Sci.* **10, 1350–1357 (2017).**

There is no SEI analysis for the SSC anode. It is not clear that how much capacity loss is due to the SEI formation for the SSC anode.

Author reply: We appreciate the reviewer #2's concern on SEI.

Our previous report (*J. Phys. Chem. C* **121**, 26155-26162 (2017)) exhibited the SEI formation on few-layer graphene-coated silicon. In this case, the cycle retention after 200 cycles is 72.1%. FEC in electrolyte improves the cycling stability. SSC in this manuscript enhances cycling stability rather than few-layer graphene-coated silicon as shown in Supplementary Fig. 1a. The capacity retention of SSC after 200 cycles is 86.6%; SSC has ICE of 86.4% higher than few-layer graphene-coated silicon (83%). Carbon surrounding silicon particles in SSC effectively avoids surface side reaction. So here we could expect lower SEI formation than the previous similar electrode.

Our new data, STEM-EELS exhibit that LiF as a component of SEI forms mainly on the carbon components of SSC (Supplementary Figs. 11 and 12). Newly added XPS spectra exhibit a tendency to increase LiF component over cycling (Supplementary Fig. 13). We have considered carefully the issue on SEI for the previous reviews. We have already discussed SEI analyses using XANES, ICP-AES, EPMA, and DC-IR in our manuscript, Figures 3b-f and Supplementary Figures 8-10, 14, 16. Especially, ICP data (Supplementary Figs. 8-10) show quantitative Li^+ loss owing to SEI formation (inactive Li) in anode.

XANES signals (Figs. 3b-f) show the growth of lithium silicate (Li_xSiO_y) on silicon in SSC. Moreover, the ratio of oxide component to silicon can be analyzed through PFY-XANES analysis. As shown in Figs. 3c and f, the spectra change little over cycling (silicon oxide : silicon = 1:9). Carbon components in SSC delay effectively the penetration of electrolyte into silicon particles.

From these results, the amount of SEI mostly formed on carbon components such as graphite and the carbon in SSC. The ratio of graphite and the carbon in SSC is 82.1 : 7.1 (=14.9 wt% (wt% of SSC in anode) \times 47.7% (carbon ratio in SSC)). During Cell B (capacity of 430 mAh) operation, the lithium loss by SEI at first and 250th cycles is 74.9 and 81.31 mAh, respectively (Supplementary Table. 1).

Then, roughly estimated lithium loss by SEI on SSC at first and 250th cycles is 6.0 and 6.5 mAh, respectively. That is about 1.4% and 1.5% of the cell capacity at first and 250th cycles, respectively.

The specific properties of SEIs on different components as well as the ratios of volumes and surfaces of materials could affect the calculation result but might not significantly change the estimation. We believe reviewer #2 also understands that direct and precise measurement of quantitative Li loss owing to SEI formation on each individual material in an anode is extremely difficult and can be further research project for deeper understanding of composite anode.

We added manuscript lines 262-264, SI lines 404-412 and Supplementary Note 10 as follows.

Manuscript line 262-264

... (initial Coulombic efficiency: 86.4%, the capacity retention at 200th cycle: 86.6%, see Supplementary Fig. 1 and Supplementary Note 9 for electrochemical properties and SEI formation on SSC, respectively).

SI line 404-412

Along with PFY-XANES result (Figs. 3c and f), this STEM-EELS results exhibit that LiF as a component of SEI forms mainly on the carbon components of SSC. The ratio of graphite and the carbon in SSC is 82.1 : 7.1 (= 14.9 wt% (wt% of SSC in anode) × 47.7% (carbon ratio in SSC)). During Cell B (capacity of 430 mAh) operation, the lithium loss by SEI at first and 250th cycles is 74.9 and 81.31 mAh, respectively (Supplementary Table 1). Then, the roughly estimated lithium losses by SEI on SSC at first and 250th cycles is 6.0 and 6.5 mAh, respectively. The specific properties of SEIs on different components as well as the ratios of volumes and surfaces of materials could affect the calculation result, but might not significantly change the estimation.

SI lines 414-437

Supplementary Note 10. X-ray photoelectron spectroscopy (XPS)

For *ex situ* X-ray photoelectron spectroscopy (XPS) measurements (PHI Quantera-II), the core-level spectra were measured using Al K α as the excitation source (1486.6 eV) at an accelerating voltage of 1 kV. All spectra were referenced to the C 1s peak at 284.8 eV. The cells were disassembled in an Ar-filled glovebox and rinsed in DMC for 5 min, followed by 30 min drying under vacuum. Subsequently, the electrodes were loaded into an in-house airtight vessel and transferred to the instrument without exposure to ambient air.

The chemical composition of the SEI in Cell B was investigated using XPS. The results are shown in Supplementary Fig. 13. The SEI surface consists of organic compounds such as ROLi (where R is a hydrogen, hydrocarbon side chain, or group of atoms) and inorganic compounds such as LiF, Li₂CO₃, and Li₂O. Supplementary Fig. 13 also exhibits a tendency for the LiF (684.8 eV), Li₂CO₃ (531.8 eV), and C–O–C (286.4 eV) components to increase with cycling. In addition, all the C 1s XPS spectra (Supplementary Fig. 13c) exhibit a peak at ~284 eV, which corresponds to the underlying graphite component of the anodes (the penetration depth of XPS is several nanometres). This suggests that the SEI layer is still thinner than the maximum XPS penetration depth of ca. 10 nm⁽²⁸⁾ after 250 cycles. Thus, the growth of the SEI was well controlled, which is in good agreement with the ICP results.

The formation of LiF is attributed to the decomposition of FEC (a component of the electrolyte). LiF forms on the outside of the SSC particles, where the coal tar pitch is situated. However, LiF is not observed inside the SSC particles, as shown in the TEM results in Supplementary Note 9. Thus, the SSC structure effectively hinders the electrolyte from penetrating into the silicon particles.

Supplementary Figure 13. Solid electrolyte interphase (SEI) chemical composition on cycled anodes of Cell B. X-ray photoelectron spectroscopy (XPS) measurements of anodes of Cell B every 50 cycles. a F 1s, b O 1s, and c C 1s spectra.

Supplementary reference

28. Louli, A. J., Ellis, L. D. & Dahn, J. R. Operando pressure measurements reveal solid electrolyte interphase growth to rank Li-ion cell performance. *Joule* **3**, 745–761 (2019).

The authors stated that the accumulation of the remaining Li causes the volume expansion of Si particles, which causes the large pressure for the graphite component, leading to the capacity degradation of graphite. But, as compared to the large volume expansion during the Si lithiation, the remaining of Li ions only cause less volume expansion. However, the authors claimed that the pressure caused by the accumulation of Li ions becomes a major player in generating the pressure for the graphite component.

Author reply: We gratefully appreciate the reviewer #2’s inspirational concern again, “as compared to the large volume expansion during the Si lithiation, the remaining of Li ions only cause less volume expansion”. That is a very good point. We believe that this answer about this comment improves our manuscript. As like reviewer #2’s comment, graphite in anode experiences the large pressure variation during a cycle. A key parameter is critical pressure rather than pressure variation during a cycle because of phase transition. Until the pressure reaches the critical pressure, staging transition of graphite does not occur. The pressure is gradually increasing owing to Li⁺ accumulation in silicon over cycling. Whenever the pressure is higher than the critical pressure, the capacity degradation of graphite accelerates owing to the pressure-induced staging transition. That is our result; experimental data as follows help in understanding our main claim.

1. The amount of accumulated Li⁺ in anode was determined by *in situ* thickness measurement (Fig. 3g) and ICP-AES (Supplementary Fig. 8); this Li⁺ accumulation arising in silicon visualized by STEM-EELS (Supplementary Fig. 12).
2. Through (de)lithiation behavior in each material and pressure evolution in graphite determined by XRD, the relationship between pressure and electrochemical behavior could be understood (Fig. 2 & Supplementary Fig. 18).
3. Our claim, pressure hinders lithiation in graphite, is well supported by the previous studies on pressure-induced staging transition (references 30-34 in manuscript). The critical pressure of HPD-graphite for staging transition would be ca. 0.33 GPa (Figs. 4a&b and Supplementary Fig. 18). This critical pressure depends on material property of graphite (Fig. 4 & Supplementary Fig. 2)

Reviewer #1 (Remarks to the Author):

The revision is good. I recommend it to be published in NC.

Reviewer #2 (Remarks to the Author):

The authors have addressed all inquiries raised by the reviewer, in particularly adding the electrochemical data of SSC and Figure S 11. There are no further comments from the reviewer. The data is very promising and hope to be used for commercializing Si anodes for lithium-ion batteries.